# EFFICIENT META REINFORCEMENT LEARNING VIA META GOAL GENERATION

## ABSTRACT

Meta reinforcement learning (meta-RL) is able to accelerate the acquisition of new tasks by learning from past experience. Current meta-RL methods usually learn to adapt to new tasks by directly optimizing the parameters of policies over primitive actions. However, for complex tasks which requires sophisticated control strategies, it would be quite inefficient to directly learn such a meta-policy. Moreover, this problem can become more severe and even fail in spare reward settings, which is quite common in practice. To this end, we propose a new meta-RL algorithm called meta goal-generation for hierarchical RL (MGHRL) by leveraging hierarchical actor-critic framework. Instead of directly generate policies over primitive actions for new tasks, MGHRL learns to generate high-level meta strategies over subgoals given past experience and leaves the rest of how to achieve subgoals as independent RL subtasks. Our empirical results on several challenging simulated robotics environments show that our method enables more efficient and effective meta-learning from past experience and outperforms state-of-the-art meta-RL and Hierarchical-RL methods in sparse reward settings.

## 1 INTRODUCTION

Deep Reinforcement Learning (DRL) has recently shown a great success on a wide range of tasks, ranging from games (Mnih et al., 2015) to robotics control (Levine et al., 2016; Bengio & LeCun, 2016). However, for more complex problems with larger state and action spaces or sparse reward settings, traditional DRL methods hardly works. Hierarchical reinforcement learning (HRL) in which multiple layers of policies are trained to learn to operate on different levels of temporal abstraction, has long held the promise to learn such difficult tasks (Dayan & Hinton, 1992; Parr & Russell, 1997; Barto & Mahadevan, 2003). By decomposing a complex problem into subproblems, HRL significantly reduces the difficulty of solving specific task. Learning multiple levels of policies in parallel is challenging due to non-stationary state transition functions. Recent HRL approaches (Nachum et al., 2018; Levy et al., 2019) use states as goals directly, allowing simple and fast training of the lower layer.

Human intelligence is remarkable for their fast adaptation to many new situations using the knowledge learned from past experience. However, agents trained by conventional DRL methods mentioned above can only learn one separate policy per task, failing to generalize to new tasks without additional large amount of training data. Meta reinforcement learning (meta-RL) addresses such problems by learning how to learn. Given a number of tasks with similar structures, meta-RL methods enable agents learn such structure from previous experience on many tasks. Thus when encountering a new task, agents can quickly adapt to it with only a small amount of experience.

Most current meta-RL methods leverage experience from previous tasks to adapt to new tasks by directly learn the policy parameters over primitive action space. (Finn et al., 2017; Rakelly et al., 2019). Such approaches suffer from two problems: (i) For complex tasks which requires sophisticated control strategies, it would be quite inefficient to directly learn such policy with one nonlinear function approximator and the adaptation to new tasks is prone to be inaccurate. This problem can become more severe in spare reward settings. (ii) When the task distribution is much wider (riding bicycle as meta-train task and riding motorcycle as meta-test task), these methods can hardly be effective since primitive action execution mechanism is entirely different although they may share

a similar high-level strategy. Moreover, existing current meta-RL methods perform badly in sparse reward settings, which are quite common in real world.

In this paper, we aim at tackling the problems mentioned above by proposing an efficient hierarchical meta-RL method that realizes meta learning high-level goal generation and leaves the learning of low-level policy for independent RL. Intuitively, this is quite similar to how a human being behaves: we usually transfer the overall understanding of similar tasks rather than remember specific actions. Our meta goal-generation framework is built on top of the architecture of PEARL (Rakelly et al., 2019) and a two level hierarchy inspired by HAC (Levy et al., 2019). Our evaluation on several simulated robotics environments (Plappert et al., 2018) shows the superiority of MGHRL to state-of-the-art meta-RL and hierarchical RL methods in sparse reward settings.

Generally, our contributions are as follows:

- We propose an algorithm that achieves efficient meta reinforcement learning on challenging robotics environments with sparse reward settings and outperforms other leading methods.

- Similar to the way humans leverage past experience to learn new complex tasks, our algorithm focuses on meta learning the overall strategy for different tasks, which provides a much simpler and better way for meta RL comparing with directly learning the detailed solution.

Since we focus on meta goal-generation and leave the low level policy for independent learning, we believe our algorithm can still accelerate the acquisition of new tasks sampled from much wider task distributions. For example, to learn tasks such as riding bicycles and riding a motorcycle, the two primitive action execution mechanism are entirely different but the two learning process still share similar high-level structures. Through meta goal-generation learning, we expect our method can still accelerate the acquisition of such tasks. We leave these for future work to explore.

## 2 RELATED WORK

Our algorithm is based on meta learning framework (Thrun & Pratt, 1998; Schmidhuber, 1987; Bengio et al., 1991), which aims to learn models that can adapt quickly to new tasks. Meta learning algorithms for few-shot supervised learning problems have explored a wide variety of approaches and architectures (Santoro et al., 2016; Vinyals et al., 2016; Ravi & Larochelle, 2017). In the context of reinforcement learning, recurrent (Duan et al., 2016; Wang et al., 2016) and recursive (Mishra et al., 2018) meta-RL methods adapt to new tasks by aggregating experience into a latent representation on which the policy is conditioned. Another set of methods is gradient-based meta reinforcement learning (Finn et al., 2017; Stadie et al., 2018; Rothfuss et al., 2019; Xu et al., 2018). Its objective is to learn an initialization such that after one or few steps of policy gradients the agent attains full performance on a new task. These methods focus on on-policy meta learning which are usually sample inefficient. Our algorithm is closely related to probabilistic embeddings for actor-critic RL (PEARL) (Rakelly et al., 2019), which is an off-policy meta RL algorithm. PEARL leverages posterior sampling to decouple the problems of inferring the task and solving it, which greatly enhances meta-learning efficiency. However, when facing complex tasks that require sophisticated control strategies, PEARL cannot effectively learn a proper meta-policy as we will show in Section 5.

Discovering meaningful and effective hierarchical policies is a longstanding research problem in RL (Dayan & Hinton, 1992; Parr & Russell, 1997; Sutton et al., 1999; Bacon et al., 2017; Dietterich, 2000). Schmidhuber 1987 proposed a HRL approach that can support multiple levels. Multi-level hierarchies have the potential to accelerate learning in sparse reward tasks because they can divide a problem into a set of short-horizon subproblems. Nachum et al. 2018 proposed HIRO, a 2-level HRL approach that can learn off-policy and outperforms two other popular HRL techniques used in continuous domains: Option-Critic (Bacon et al., 2017) and FeUdal Networks (Vezhnevets et al., 2017). Our algorithm is built on Hierarchical actor-critic (Levy et al., 2019), which is a framework that can learn multiple levels of policies in parallel. Most current HRL works focus on the learning problem in a single task and few of them considers to take advantage of HRL for multi-task or meta-learning tasks. MLSH (Frans et al.) is such a work which also combines meta-RL with Hierarchical RL. It focuses on meta learning on the low level policy and need to retrain its high level policy when facing new tasks. In contrast, with the key insight that humans leverage abstracted prior knowledge

obtained from past experience, our method focus on meta learning high level overall strategy using past experience and leave the detailed action execution for independent RL.

# 3 BACKGROUND

## 3.1 META REINFORCEMENT LEARNING

In our meta learning scenario, we assume a distribution of tasks $p(\tau)$ that we want our model to adapt to. Each task correspond to a different Markov decision process (MDP), $M_i = \{S, A, T_i, R_i\}$, with state space $S$, action space $A$, transition distribution $T_i$, and reward function $R_i$. We assume that the transitions and reward function vary across tasks. Meta RL aims to learn a policy that can adapt to maximize the expected reward for novel tasks from $p(\tau)$ as efficiently as possible.

PEARL (Rakelly et al., 2019) is an off-policy meta-reinforcement learning method that drastically improves sample efficiency comparing to previous meta-RL algorithms. The meta-training process of PEARL learns a policy that adapts to the task at hand by conditioning the history of past transitions, which we refer to as context $c$. Specifically, for the $i$th transition in task $\tau$, $c_i^\tau = (s_i, a_i, r_i, s_i')$. PEARL leverages an inference network $q_\phi(z|c)$ and outputs probabilistic latent variable $z$. The parameters of $q(z|c)$ are optimized jointly with the parameters of the actor $\pi_\theta(a|s, z)$ and critic $Q_\theta(s, a, z)$, using the reparametrization trick (Kingma & Welling, 2014) to compute gradients for parameters of $q_\phi(z|c)$ through sampled $z$'s.

## 3.2 HIERARCHICAL ACTOR-CRITIC

HAC (Levy et al., 2019) aims to accelerate learning by enabling hierarchical agents to jointly learn a multi-level hierarchy of policies in parallel. HAC is comprised of two components: a particular hierarchical architecture and a method for learning the multiple levels of policies in parallel given sparse rewards. The hierarchies produced by HAC have a specific architecture consisting of a set of nested, goal-conditioned policies that use the state space as the mechanism for breaking down a task into subtasks. HAC extends the idea of Hindsight Experience Replay (Andrychowicz et al., 2017) by creating two types of hindsight transitions. Hindsight action transition simulates a transition function that uses the optimal low level policy while hindsight goal transition use the final states achieved as the goal state in each step's transition. They enable agents to learn multiple policies in parallel using only sparse reward functions.

# 4 ALGORITHM

## 4.1 TWO-LEVEL HIERARCHY

We set up a hierarchical two-layer RL structure similar to HAC. The high level network uses policy $\mu^h$ to generate goals for temporally extended periods in terms of desired observations. In our task they correspond to the positional features of the gripper. The low level policy $\mu^l$ directly controls the agent and produces actions for moving towards the desired goals.

As shown in Figure 1 (a), the high level policy $\mu^h$ observes the state and produces a high level action (or goal) $g^t$. Low level policy $\mu^l$ has at most K attempts of primitive action to achieve $g^t$. Here, K which can be viewed as the maximum horizon of a subgoal action is a hyperparameter given by the user. As long as the low level policy $\mu^l$ run out of K attempts or $g^t$ is achieved, this high level transition terminates. The high level policy uses agent's current state as the new observation and produced another goal for low level policy to achieve.

We use an intrinsic reward function in which a reward of 0 is granted only if the goal produced by high level policy is achieved and a reward of -1 otherwise. Note that the environment's return (i.e. whether the agent successfully accomplished the task) will not affect the reward received by the low level policy. In our evaluation on simulated robotics environments, we use the positional features of the observations as the representation for $g^t$. A goal $g^t$ is judged to be achieved only if the distance between $g^t$ and the gripper's current position $s_{n+1}$ is less than threshold $l$.

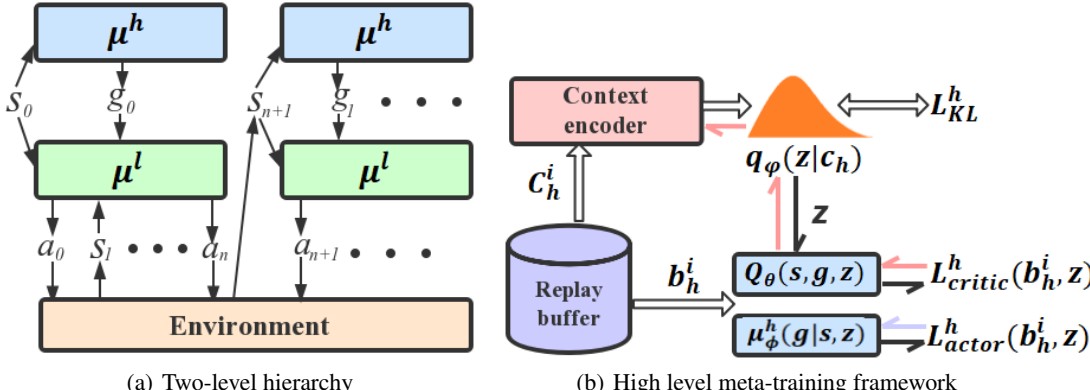

(a) Two-level hierarchy        (b) High level meta-training framework

Figure 1: (a) **Two level hierarchy of MGHRL**. High level policy $\mu^h$ takes in state and outputs goals at intervals. Low level policy $\mu^l$ takes in state and desired goals to generate primitive actions. (b) **High level meta-training framework of MGHRL**. The context encoder network uses high-level context data $C_h^i$ to infer the posterior over the latent context variable z, and is optimized with gradients from the critic as well as from an information bottleneck on z. The actor network $\mu_\phi^h(g|s,z)$ and critic network $Q_\theta(s,g,z)$ both treat z as part of the state.

## 4.2 META GOAL-GENERATION FOR HIERARCHICAL REINFORCEMENT LEARNING

One primary motivation for our hierarchical meta reinforcement learning strategy is that, when people try to solve new tasks using prior experience, they usually focus on the overall strategy we used in previous tasks instead of the primitive action execution mechanism. Most state-of-the-art meta learning methods (Rakelly et al., 2019; Finn et al., 2017) leverage experiences from previous tasks to quickly adapt to the new tasks and directly learn the policy parameters. However, it can be difficult to meta learn a proper policy that consider both the overall strategy and detailed action execution in some complex tasks. Using only one level of non-linear function approximator may lead the agents to learning an inaccurate meta-policy when both the overall structure and primitive action execution mechanism are complex. Moreover, in sparse reward settings which is a common situation in real-world problems, current meta learning algorithms do not perform well enough since their training methods are based on non-hierarchical RL methods like TRPO (Schulman et al., 2015), SAC (Haarnoja et al., 2018), etc. These methods suffer from the difficulty of effective exploration and the lack of positive update signals.

To address the problem mentioned above, we take advantage of our two-level hierarchy structure and propose a new meta reinforcement learning framework called meta goal-generation for hierarchical RL (MGHRL). Instead of learning to generate detailed strategy for new tasks, MGHRL learns to generate overall strategy (goals) given past experience and leaves the detailed method of how to achieve the goals for independent RL. We leverage PEARL framework (Rakelly et al., 2019) and independently train a high level meta-policy which is able to quickly adapt to new tasks and generate proper goals. Note that off-policy RL method is indispensable in our structure when training high level policy due to its excellent sample efficiency during meta-training. Good sample efficiency enables fast adaptation by accumulating experience online, and performs structured exploration by reasoning about uncertainty over tasks, which is crucial to hierarchical parallel training framework. We leave the low level policy to be trained independently with non-meta RL algorithm using hindsight experience replay mechanism. In our simulated robotics experiments, the low level policy aims to move the gripper to the desired goal position which can be reused when switching to other tasks. Thus we only need to train a single set of low-level polices which can be shared and reused across different tasks. On the other hand, in other situations where the tasks are from different domains, for example, when we use our experience of learning riding bicycle to help us learn how to ride a motorcycle, the primitive action execution mechanism are entirely different. In this case, we can train low level policy independently on new tasks without using past experience. Our main insight is that when dealing with entirely new tasks, the primitive action execution mechanism can be entirely different but the general strategy of how to accomplish the new tasks and prior tasks can be similar.

With meta learning on high level policy, our algorithm still greatly accelerate the acquisition of new tasks.

Our high level meta-RL network uses a probabilistic embedding actor-critic framework similar to PEARL. The network consists of two parts. The first part is a context encoder which leverages data from a variety of training tasks to learn to infer the value of $z$ from a recent history of high-level experience in the new task, where $z$ functions as a latent probabilistic context variable. The encoder network parameterized by $\varphi$ takes context (experience) $c_h$ as input and output posterior $q_\varphi(z|c_h)$ as a permutation-invariant function (Rakelly et al., 2019) of prior high level experience. The context $c_h$ consists of experience $\{s, g, r, s'\}$ collected using hindsight technique as we will introduce in Section 4.3. Then we can sample $z$ from the posterior and compute policy output and $Q$ value conditioned on it. Through posterior sampling via latent contexts, the high level network can learn to infer new tasks efficiently using past experience. The second part is built on top of soft actor-critic algorithm (Haarnoja et al., 2018). As we mentioned before, samples from the posterior belief are passed to actor $\mu_\phi^h(g|s, z)$ and critic $Q_\theta(s, g, z)$ to make predictions of the sampled task. Note that we treat $z$ as part of the state when we implement with SAC.

The actor and critic are trained to predict optimally given $z$ with batches of transitions drawn uniformly from the entire replay buffer. The context encoder is optimized using gradients from the critic. We summarize our meta-training procedure in Algorithm1 and Figure 1 (b). Concretely, for each training task drawn from task distribution, we sample context and generate hindsight transitions for both levels of hierarchy ($line\ 4 \sim 13$) by performing current policy. Then we train high level and low level networks with the collected data ($line\ 16 \sim 22$).

---

**Algorithm 1** MGHRL Meta-training

---

**Require:** Batch of training tasks $\{\tau_i\}_{i=1,...,T}$ from $p(\tau)$, maximum horizon $K$ of subgoal action
  1: Initialize replay buffers $\mathcal{B}_h^i, \mathcal{B}_l^i$ for each training task
  2: **while** not done **do**
  3:    **for** each task $\tau_i$ **do**
  4:       Initialize high-level context $c_h^i = \{\}$
  5:       **for** m=1,...,M **do**
  6:          Sample $z \sim q_\varphi(z|c_h^i)$
  7:          $g_i \leftarrow \pi_h(g|s, z)$
  8:          **for** $K$ attempts or until $g_i$ achieved **do**
  9:             Gather data using $a_i \leftarrow \pi_l(a|s, g)$
10:             Generate hindsight action transition, hindsight goal transition and add to $\mathcal{B}_l^i$
11:          **end for**
12:          Generate hindsight transitions, subgoal test transitions and add to $\mathcal{B}_h^i$
13:          Sample high level context $c_h^i = \{s_j, g_j, r_j, s_j'\}_{j=1,...,N} \sim \mathcal{B}_h^i$
14:       **end for**
15:    **end for**
16:    **for** steps in training steps **do**
17:       **for** each task $\tau_i$ **do**
18:          Sample high level context $c_h^i \sim \mathcal{B}_h^i$ and RL batch $b_h^i \sim \mathcal{B}_h^i, b_l^i \sim \mathcal{B}_l^i$
19:          Sample $z \sim q_\varphi(z|c_h^i)$ and calculate $L_{actor}^h(b_h^i, z), L_{critic}^h(b_h^i, z), L_{KL}^h$
20:          Update low level actor and critic network with $b_l^i$
21:       **end for**
22:       Update high level networks with $\sum_i L_{actor}^h, \sum_i L_{critic}^h, \sum_i L_{KL}^h$
23:    **end for**
24: **end while**

---

## 4.3 Parallel training strategy

Efficient meta reinforcement learning requires parallel training for the two levels of our networks. To achieve parallel training paradigm, there exists two main issues in MGHRL framework. The first issue for meta learning hierarchies is that agents need to act randomly to reach their goals and obtain the sparse reward which proves to be quite difficult for both levels. We need other strategies to ensure

each level learn effectively in sparse reward settings. The second issue is the non-stationary problem when we do parallel training for the high level and low level networks. Whenever low level policy $\pi^l$ changes, the high level transition function is likely to change as well. Old off-policy experience may exhibit different transitions conditioned on the same goals, making the transition invalid for training. The same problem occurs when the low level is exploring with some random noise. Thus in our algorithm, we rewrite the past experience transitions as hindsight action transitions (Andrychowicz et al., 2017), and supplement both levels of hierarchy with additional sets of transitions as was done in HAC.

Hindsight action transition simulates a transition function that uses the optimal low level policy which enables our framework to train both levels in parallel. It substitutes the action component in high level transition to the next state achieved in low level. If the original high level transition is $[s_t, g_t, r_t, s_{t+1}]$, the hindsight action transition will be $[s_t, s_{t+1}^g, r_t, s_{t+1}]$, where $s_{t+1}^g$ represents the component vector of next state that matches the goal vector. The new transition we get is independent of changing or exploring low level policy since it's always optimal.

We utilize hindsight goal transition and subgoal test transition to further address the problems mentioned before. Hindsight goal transition is created for both levels. After at most $K$ attempts is executed, the final states achieved is used as the goal state in each step's transition instead of the original goal state. And the reward will be updated to reflect the new goal state. Subgoal test transition is meant to compensate for the drawbacks brought by hindsight action transition. Hindsight action transition prefer the shortest path of goals that have been found but may ignore the range of goals that the low level policy is able to reach. Thus, subgoal test transition add a penalty of $-K$ to the reward if the goal is not achieved after $K$ attempts by low level policy and set the discount rate to $0$ to avoid non-stationary issues.

## 5 Experiments

We evaluated our algorithm on several challenging continuous control robotics tasks (integrated with OpenAI Gym) (Plappert et al., 2018), simulated via the MuJoCo physics simulator (Todorov et al., 2012). Visualizations of these environments are shown in Figure 2. More details on each environment can be found at `https://openai.com/blog/ingredients-for-robotics-research/`.

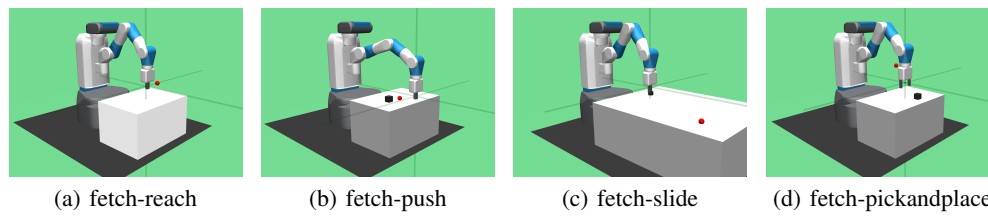

| (a) fetch-reach | (b) fetch-push | (c) fetch-slide | (d) fetch-pickandplace |

Figure 2: The four evaluated robotics environments

**Fetch-Reach** Fetch has to move the gripper to the desired goal position. This task is very easy to learn and is therefore a suitable benchmark to ensure that a new idea works at all.

**Fetch-Push** A box is placed on a table in front of the robot and Fetch has to move a box by pushing it until it reaches a desired goal position. The robot fingers are locked to prevent grasping. The learned behavior is usually a mixture of pushing and rolling.

**Fetch-Slide** A puck is placed on a long slippery table and the target position is outside of the robots reach so Fetch has to hit the puck across a long table such that it slides and comes to rest on the desired goal.

**Fetch-PickandPlace** Fetch has to pick up a box from a table using its gripper and move it to a desired goal located on the table.

## 5.1 ENVIRONMENTAL SETUP

In all our experiments, we compare our algorithm to baselines including PEARL with dense reward, PEARL with sparse reward and HAC with shared policy. The last one means we train a shared HAC policy jointly across all meta-train tasks sampled from the whole task distribution. Note that Rakelly et al. (2019) has already proved that PEARL greatly outperforms other existing meta RL methods like MAML (Finn et al., 2017), ProMP (Rothfuss et al., 2019) at both sample efficiency and final performance. Thus we mainly compare our results with PEARL using its public source code. In addition, for a fair comparison, we modify the HAC source code with SAC algorithm which are considered to be much powerful than DDPG in the original implementation (Haarnoja et al., 2018), to ensure the consistence to PEARL and MGHRL.

We set the goal space to be the set of all possible positions of the gripper, in which a goal is a 3-d vector. In the environments, the low level policy of our algorithm aims to move the gripper to the desired goal position. Such policy won't change at all when switching to other tasks since the mechanism of moving gripper keeps the same between different tasks. Thus we use a shared policy trained jointly across all tasks for the low level of MGHRL. In all four scenarios, we set the maximum low-level horizon $K$ to be 10 and the distance threshold to be 0.05. The high level context data sampler $S_c^h$ samples uniformly from the most recently collected batch of data, which is recollected every 1000 meta-training steps. Unlike HAC, we use target networks for both levels, which updates with $\tau = 0.005$. All context encoder, actor and critic neural networks had three hidden layers, with 300 nodes in each layer. The discount factor was set to $\gamma = 0.99$. We use a sparse reward function in which a reward of 0 is granted only if the terminal goal given by the environment is achieved and a reward of -1 otherwise. The dense reward used in our baseline is a value corresponding to the distance between current position of the box (or gripper in fetch-reach case) and desired goal position. In all four scenarios, we do our experiments on 50 train tasks and 10 test tasks, where the difference between each task is in the terminal goal position we want the box or gripper to reach.

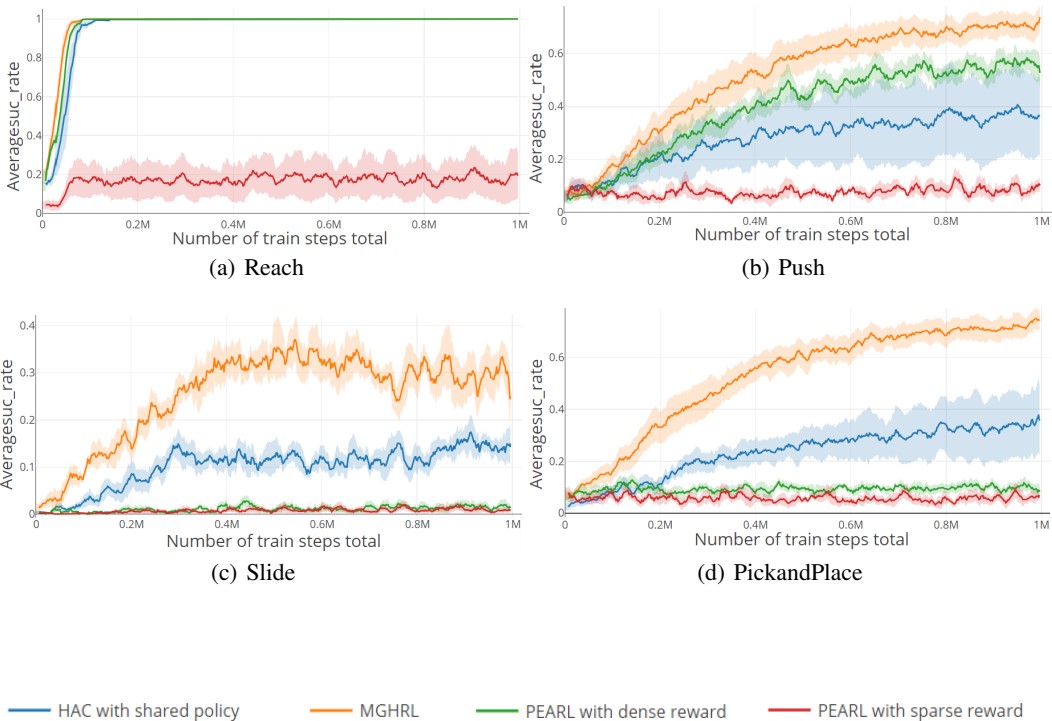

Figure 3: Average success rates for MGHRL, HAC, PEARL agents in each task, each algorithm was trained for 1e6 steps. The error bar shows 1 standard deviation.

## 5.2 RESULTS

We evaluate the performance of approaches in the term of the average success rate. As shown in Figure 3, in Fetch-reach environment which is very easy to learn as we mentioned before, the tested methods except PEARL with sparse reward all reach a final performance of $100\%$ success rate. In other three scenarios, MGHRL significantly outperforms the other three method in such sparse reward settings. Our two-level hierarchy and hindsight transitions significantly decrease the difficulty of meta learning with sparse reward. As we expected, PEARL performs badly in sparse reward settings. The original version of PEARL is based on SAC, such non-hierarchical RL method has been proved to perform badly before on challenging tasks with sparse reward settings. Thus it is reasonable that PEARL, which can be viewed as a meta-version of SAC, performs badly as well in sparse reward settings. HAC with shared policy generally performs better than PEARL in Fetch-slide and Fetch-pickandplace environments. We assume that it is because in our settings since we only change the terminal goals' positions to create different tasks, thus it is possible that the policy learned from one task will work on other task whose terminal goal positions are very close to previous training ones. But such method lacks generalization ability and cannot always achieve good performance when tested on varied tasks as shown in our results.

We also compare our method to PEARL with dense reward to demonstrate that MGHRL is able to more efficiently and accurately meta learn from past experience. Shown in Figure 3, generally, our algorithm still outperforms PEARL and adapts to new tasks much more quickly. In such environments with sophisticated control strategies, directly using PEARL to meta learn a policy that consider both overall strategy and detailed execution would decrease prediction accuracy and sample efficiency. Thus it is better to decompose the meta-RL training process and focus on meta goal-generation learning. Moreover, under dense reward settings of these challenging tasks, the critic of PEARL has to approximate a highly non-linear function that includes the Euclidean distance between positions and the difference between two quaternions for rotations (Plappert et al., 2018). As our method use a hierarchical structure, learning with the sparse return is much simpler since the critic only has to differentiate between successful and failed states.

## 6 CONCLUSION

In this paper, we have presented a hierarchical meta-RL algorithm, MGHRL, which realizes meta goal-generation and leave the low-level policy for independent RL. MGHRL aims to more efficiently and accurately meta learn from past experience by focusing on learning the overall strategy of tasks instead of learning detailed action execution. Our experimental results on a range of simulated robotics environments show the superiority of MGHRL over state-of-the-art meta RL and hierarchical RL methods in challenging and practical sparse reward settings.

We believe our work open up many directions in training agents that can quickly adapt to new tasks sampled from much wider distribution efficiently. Currently, we have only conducted experiments on meta learning tasks with relatively narrow task distribution (e.g. different goal positions of the box). As future work, we expect our algorithm can accelerate the acquisition of entirely new tasks (i.e. using fetch-push and fetch-slide as meta train tasks and using fetch-pickandplace as meta test task) by only meta learning overall strategy and leaving the details of primitive action execution mechanism for further separate low-level policy learning. Moreover, we note our results on some tasks are still far from perfect. There is still much work left for future research to improve meta-RL methods' performance on those tasks.

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
