# OpenReview forum: "Efficient meta reinforcement learning via meta goal generation"
_ICLR.cc/2020/Conference — Reject_

### Official Review · AnonReviewer1 · 2019-10-20
**Official Blind Review #1**

**Rating:** 1

**Review:**

This paper studies the problem of leveraging past experience to quickly solve new control tasks. The starting point (and perhaps the main contribution) is the observation that some tasks have similar high-level goals, while differing in how those goals are achieved. To that end, the paper introduces an meta-RL algorithm that, given a new task, attempts to solve it by adapting a high-level, goal-setting module, and learn a new, low-level policy to reach each commanded goal. The proposed method might be viewed as a combination of PEARL [Rakelly 19] and HAC [Levy 19]. The proposed method is compared against state-of-the-art hierarchical RL and meta-RL methods on four robotic manipulation tasks. The proposed method outperforms the baselines on each task.

While the proposed method is quite strong empirically, I am leaning towards rejecting this paper because many of the claims made in the paper are not empirically validated. While much emphasis is put on the hierarchical aspect of the algorithm, I don't think that the tasks used in the experiments require hierarchy to solve (see [Plappert 18]). In the introduction, the second claim is that the proposed method "focus[es] on meta learning the overall strategy … [and] provides a simpler and better way for meta RL." While the experiments show that the proposed method learns better than baselines, I don't think the paper show that the proposed method learns some sort of "overall strategy." I don't think that the proposed method is simpler than the baselines.

A second concern is that I'm confused about the experimental protocol. If the high-level policy outputs a desired XYZ position for the gripper (Section 5.1), how can the high-level policy indicate when the gripper should be closed to pick up the block? How is the reward function for the low-level policy defined? PEARL doesn't have access to this extra information (the reward function), right?

A third concern is the large number of grammatical errors in the paper.

I would consider increasing my review if (1) a new plot were added to visualize the commanded subgoals (I have a hunch that the high-level policy directly outputs the true goal, obviating the need for hierarchy and contradicting the claim that the method "learns to generate high-level meta-strategies over subgoals"); (2) the experimental protocol were clarified; and (3) the number of grammatical errors were significantly reduced.

Other comments:
* "inefficient to to directly learn such a meta policy" -- Why? Also, "to to" is repeated.
* "Deep Reinforcement learning" -- "Reinforcement" shouldn't be capitalized.
* "failing to generalize" -- Can you add a citation?
* "it would be quite inefficient to directly learn such a policy…": Doesn't [Plappert 18] do exactly this?
* "When the tasks distribution is much wider … these methods can hardly be effective…" -- Where is this claim substantiated? Also, "tasks" should be singular.
* "sparse reward settings which is" -> "sparse reward settings, which are"
* "the above mentioned problems" -> "the problems mentioned above"
* "our algorithm focus on meta learning … which provides a much simpler …" -- Where is it shown that the proposed method is simpler? Also, "focus" should be "focuses"
* "1991),which" -- Missing space
* "complex tasks which requires" -> "complex tasks that require"
* "Nachum et al … set of sub-policies" -- Run on sentence.
* "... human leverage…" -- Don't humans also transfer low-level knowledge across tasks, in addition to high-level knowledge? Also, "human" should be plural.
* "algorithms.The" -- Missing a space, I think.
* "PEARL leverages … latent variable Z" -- This sentence doesn't make sense as written.
* "z's" -- This should not be a possessive.
* "Good sample efficiency enables fast adaptation … and performs structured exploration …" -- Isn't the first part true by definition? Why does good sample efficiency perform structured exploration?
* "a goal is a 3-d vector" -- If the goal output by the high-level policy is the XYZ coordinates of the
* "SAC, such non-hierarchical" -- Grammar doesn't make sense here.
* "such non-hierarchical RL method has been proved to perform badly before on …" -- Can you add a citation? Generally, "proved" is reserved for mathematical proofs.
* "In this paper, We have" -- "We" should not be capitalized.

------------------------ UPDATE AFTER AUTHOR RESPONSE ------------------
I thank the authors for at least reading the reviews. My concerns with experiments and clarify remain unaddressed, and are amplified by reading the other reviews. I therefore vote to "reject" this paper.

**Experience Assessment:**

I have published one or two papers in this area.

**Review Assessment: Checking Correctness Of Derivations And Theory:**

N/A

**Review Assessment: Checking Correctness Of Experiments:**

I assessed the sensibility of the experiments.

**Review Assessment: Thoroughness In Paper Reading:**

I read the paper at least twice and used my best judgement in assessing the paper.

---

> ### Author Response · Authors · 2019-11-14
> **Thank you for carefully reviewing the paper**
>
> We appreciate the reviewer's valuable and constructive reviews. We will improve our paper as suggested.

---

### Official Review · AnonReviewer3 · 2019-10-24
**Official Blind Review #3**

**Rating:** 3

**Review:**

### Summary ###

In this paper, the authors focus on the problem of meta-reinforcement learning (meta-RL). Specifically, the authors consider the setting of meta-RL for goal reaching tasks where each task corresponds to an unknown goal. Existing meta-RL algorithms directly train for a policy that output low level actions, which might be inefficient in this goal-reaching setting. In this paper, the authors combine the hierarchical RL framework of HAC[1] with the probabilistic task context inference method of PEARL[2], and propose the meta-goal generation for hierarchical RL (MGHRL) algorithm. In this algorithm, a two layer hierarchical policy is used where the high level policy generate goals for the low level goal-reaching policy to reach. In order to adapt to an unknown goal, the high level policy is conditioned on the output of a task inference module to generate goals for the unknown ground truth goal. The goal-reaching policy would then use the generated goal to interact with the environment.

The authors evaluated the proposed method on simulated robotic manipulation tasks and compare to PEARL as baseline. The experiment results show that the proposed method outperforms the baseline method significantly, especially under sparse reward settings.


### Review ###

Overall I think this paper presents an interesting idea in learning fast adapting goal-reaching policies. The idea is very well presented and authors include many empirical evidence to support the proposed method. However I do find a number of shortcomings that need to be addressed.

Pro:

1. The idea for this paper is really well presented. The structure of the paper is well organized and  the authors include informative illustration to explain the architecture of the hierarchy of policies. The experiment results are easy to interpret.

2.  The authors provide a detailed description of the configurations and the hyperparameters for each experiments. Such description would be very helpful if the results in this paper are to be reproduced.

Con:

1. The experiments presented in this paper do not include appropriate comparisons to baseline methods. While indeed the proposed method outperforms PEARL, this comparison is inherently unfair. PEARL is a general meta-RL algorithm, which can adapt to arbitrary variations of reward functions and dynamics in the distribution of tasks. The proposed method only applies in the setting of goal reaching meta-RL, where each task corresponds to an unknown goal. With this information artificially encoded into the hierarchical architecture, the proposed method should certainly perform better than any general meta-RL algorithms. Therefore, directly comparing the proposed method to any general meta-RL algorithm is unfair. Instead, the authors could compare with baseline methods with builtin goal-reaching components, such as the following one: train a goal reaching policy using HER[3], and then meta-train a goal conditioned reward function using standard supervised meta-learning methods. At test time, find the goal that maximizes the adapted reward function, and then feed that goal into the HER policy for evaluation. Note that this baseline is different from the proposed method in the way that the goal reaching and goal inference were done separately both using existing methods.

2. I’m not convinced about the novelty of this paper. The proposed method seems like a straightforward combination of HAC and PEARL, and it seems to me that the two methods are combined in order to apply an existing meta-RL algorithm in a goal reaching setting rather than to create a better general meta-RL algorithm.

The idea in the paper is well presented and carefully investigated. However, I am still not convinced about the novelty of the proposed idea and the magnitude of performance improvement given the lack of proper baselines. Therefore, I would not recommend acceptance before these problems are addressed.

References

[1] Levy, Andrew, et al. "Learning multi-level hierarchies with hindsight." (2018).

[2] Rakelly, Kate, et al. "Efficient off-policy meta-reinforcement learning via probabilistic context variables." arXiv preprint arXiv:1903.08254 (2019).

[3] Andrychowicz, Marcin, et al. "Hindsight experience replay." Advances in Neural Information Processing Systems. 2017.


**Experience Assessment:**

I have published in this field for several years.

**Review Assessment: Checking Correctness Of Derivations And Theory:**

N/A

**Review Assessment: Checking Correctness Of Experiments:**

I carefully checked the experiments.

**Review Assessment: Thoroughness In Paper Reading:**

I read the paper thoroughly.

---

> ### Author Response · Authors · 2019-11-14
> **Thank you for carefully reviewing the paper**
>
> We appreciate the reviewer's valuable and constructive reviews. We will improve our paper as suggested.

---

### Official Review · AnonReviewer2 · 2019-10-25
**Official Blind Review #2**

**Rating:** 1

**Review:**

Update 11/21
I maintain my score. I like the idea and hope the authors improve the paper and submit to a future conference.

Summary
This paper combines hierarchical RL with meta-learning. The idea is that high-level plans transfer across settings (e.g. picking up a mug), while low-level execution may differ across tasks (e.g. different robot morphologies). To this end, the approach meta-learns a two-level hierarchical policy. The higher level policy conditions on a latent task context to produce high-level actions, or goals for the lower level policy. The lower level policy is trained via HER to reach these goals (it may need to be completely re-trained at test time).

Concerns and Questions
I am very concerned about the experimental results. I do not think that these tasks require hierarchy to solve, as the exact same tasks (with the same simulated robot) were solved in Hindsight Experience Replay, Andrychowicz al. 2017. Thus HER (preferably implemented with SAC rather than DDPG for fair comparison) is a vital baseline that is missing from Figure 3. Could the authors please address this point?
While the introduction and Section 4 claim that one important benefit of such a hierarchical approach is that one could transfer to more disparate tasks, there are no experiments supporting this idea. I think the addition of these experiments would greatly strengthen the paper.
In Figure 3, does “PEARL with sparse reward” refer to only the encoder receiving sparse rewards or also the actor-critic?

Writing suggestions
A suggestion about the title: consider including the word “hierarchical”
In some places the writing is quite informal, I suggest revising it (in the intro: “DRL barely works”.
I disagree with the sentence in the intro “Intuitively this is quite similar to how a human behaves”, which is said in support of the idea of transferring high-level goals instead of low-level execution. Human behavior also supports the opposite view - that we reuse primitive motions over and over in support of new goals. So I think it’s best to avoid the appeal to human behavior here (as well as in related work).
The first paragraph of section 4.2 is redundant and can be removed, or at least moved to the beginning of Section 4.

In conclusion, my current impression is that while the idea is interesting, the results achieve the same performance as a non-hierarchical method, which is not included as a baseline.


**Experience Assessment:**

I have published one or two papers in this area.

**Review Assessment: Checking Correctness Of Derivations And Theory:**

N/A

**Review Assessment: Checking Correctness Of Experiments:**

I carefully checked the experiments.

**Review Assessment: Thoroughness In Paper Reading:**

I read the paper thoroughly.

---

> ### Author Response · Authors · 2019-11-14
> **Thank you for carefully reviewing the paper**
>
> We appreciate the reviewer's valuable and constructive reviews. We will improve our paper as suggested.

---

### Decision · Program_Chairs · 2019-12-19

**Decision:**

Reject

**Comment:**

This paper combines PEARL with HAC to create a hierarchical meta-RL algorithm that operates on goals at the high level and learns low-level policies to reach those goals. Reviewers remarked that it’s well-presented and well-organized, with enough details to be mostly reproducible. In the experiments conducted, it appears to show strong results.

However there was strong consensus on two major weaknesses that render this paper unpublishable in its current form: 1) the continuous control tasks used don’t seem to require hierarchy, and 2) the baselines don’t appear to be appropriate. Reviewers remarked that a vital missing baseline is HER, and that it’s unfair to compare to PEARL, which is a more general meta-RL algorithm. The authors don’t appear to have made revisions in response to these concerns.

All reviewers made useful and constructive comments, and I urge the authors to take them into consideration when revising for a future submission.